# Treatment of Dyslipidemia through Targeted Therapy of Gut Microbiota

**DOI:** 10.3390/nu15010228

**Published:** 2023-01-02

**Authors:** Brandon Flaig, Rachel Garza, Bhavdeep Singh, Sevag Hamamah, Mihai Covasa

**Affiliations:** 1Department of Basic Medical Sciences, College of Osteopathic Medicine, Western University of Health Sciences, Pomona, CA 91766, USA; 2Department of Medicine and Biomedical Sciences, College of Medicine and Biological Science, University of Suceava, 7200229 Suceava, Romania

**Keywords:** mediterranean diet, prebiotics, probiotics, synbiotics, folate, fecal microbiota transplantation, *Akkermansia muciniphila*, *Fecalibacterium prausnitszii*

## Abstract

Dyslipidemia is a multifaceted condition with various genetic and environmental factors contributing to its pathogenesis. Further, this condition represents an important risk factor for its related sequalae including cardiovascular diseases (CVD) such as coronary artery disease (CAD) and stroke. Emerging evidence has shown that gut microbiota and their metabolites can worsen or protect against the development of dyslipidemia. Although there are currently numerous treatment modalities available including lifestyle modification and pharmacologic interventions, there has been promising research on dyslipidemia that involves the benefits of modulating gut microbiota in treating alterations in lipid metabolism. In this review, we examine the relationship between gut microbiota and dyslipidemia, the impact of gut microbiota metabolites on the development of dyslipidemia, and the current research on dietary interventions, prebiotics, probiotics, synbiotics and microbiota transplant as therapeutic modalities in prevention of cardiovascular disease. Overall, understanding the mechanisms by which gut microbiota and their metabolites affect dyslipidemia progression will help develop more precise therapeutic targets to optimize lipid metabolism.

## 1. Introduction

The human gastrointestinal (GI) tract is host to trillions of bacteria, viruses, fungi and archaea collectively termed gut microbiota, with the colon containing the greatest density of these microbes, estimated at 3.2 × 10^11^ cells per gram content [1]. Composition of gut microbiota varies between individuals due to the interaction between genetics and environmental factors. The typical human gut microbiota contains many different bacterial species, with a predominance of two phyla including a Gram-positive phylum *Firmicutes* and a Gram-negative phylum *Bacteroidetes* [2]. Other less abundant gut bacteria species belong to the phyla *Actinobacteria*, *Proteobacteria*, and *Verrucomicrobia* [3]. In a healthy gut, homeostatic microbial composition is achieved via cross talk between host and microbiota through a commensal relationship [4], though negative changes in gut microbial composition and diversity can contribute to the development of disease states [5,6,7,8,9]. Overall, gut microbiota serve a variety of host functions including regulation of metabolism and immune system as well as maintaining gut mucosal barrier integrity [10]. Further, gut microbiota contain digestive enzymes otherwise not present in the human GI tract, allowing absorption of indigestible nutrients in the intestines. More specifically, these enzymes yield important metabolites, namely short-chain fatty acids (SFCA), including butyrate, acetate, and propionate [11], shown to exert many of the beneficial effects of gut microbiota [12]. Additionally, gut microbiota also serve important roles in modifying bile acids via deconjugation, dehydroxylation, and epimerization into secondary bile acids [13]. These microbiota-derived alterations in bile acids are implicated in inflammatory signaling, intestinal immunity as well as maintaining effective lipid metabolism [14]. 

While the gut microbiota exert a myriad of beneficial effects on its host, imbalance or dysbiosis may leave the host vulnerable to pathogenic disturbances. Dysbiosis is characterized by a loss of beneficial organisms, increased growth of harmful organisms, or loss of microbial diversity and is largely influenced by environmental factors [15]. For example, the Western diet confers negative changes in gut microbiota to produce a chronic low-grade inflammation and increased intestinal barrier permeability [16]. Further, the unfavorable dietary substrates and excess energy associated with a Western diet led to increase adipocytes storage in the form of triglycerides (TG) [17]. These dietary practices coupled with physical inactivity, poor sleep, and chronic psychological stress leading to dysregulations of lipid metabolism, are major contributors of the onset of metabolic disease [18]. As such, the relationship between dysbiosis-related changes in gut microbiota and its correlation with dyslipidemia have been extensively studied in recent years with targeted gut microbiota interventions as a potential therapeutic intervention. In this review, we first explore the associations between gut microbiota dysbiosis and altered lipid metabolism. Next, we present the effects of key gut microbial metabolites on the development and progression of dyslipidemia. Then, we describe how diet impacts changes in the gut microbiota and the resulting influences on lipid metabolism. Lastly, we discuss the recent advances on targeted microbiota therapy using prebiotics, probiotics, synbiotics, and microbiota transplant for management of dyslipidemia.

## 2. Dyslipidemia and Gut Microbiota

Dyslipidemia, which broadly describes abnormal serum lipid levels, with clinical sequelae being a major culprit responsible for increasing rates of lifetime morbidity and mortality worldwide [19]. Normally, dyslipidemia is detected with a rapid lipid panel that includes values of TG, total cholesterol (TC), high density lipoprotein (HDL), low density lipoprotein (LDL), and non-HDL [20]. Dyslipidemia with elevated LDL and TG levels has been associated with a greater risk of atherosclerotic cardiovascular diseases, such as myocardial infarction and ischemic stroke [21]. Both dyslipidemias of familial and lifestyle cause have been demonstrated to have similar lipid profiles and risks of cardiovascular disease [22]. The etiology of dyslipidemia is multifaceted, with influences from genetic, lifestyle, age, environmental factors and gut dysbiosis. Several studies have demonstrated that microbiota composition profile influences cholesterol metabolism and markers of dyslipidemia. For example, colonization of mice with microbiota from donors with dyslipidemia resulted in increased intestinal cholesterol absorption and a hypercholesterolemic phenotype when compared to mice colonized with normal microbiota [23]. Similarly in humans, washed microbiota treatment has been shown to reduce serum cholesterol in dyslipidemia patients [24]. Further, using germ-free animal models, it has been shown that the effects of dietary fats on the host physiology and metabolism are dependent on the gut microbiota. For example, GF mice have decreased levels of serum triglycerides and LDL, while also being resistant to diet-induced obesity secondary to impairments in lipid metabolism within the small intestine [25]. Conventionalization with specific bacterial strains such as *Lactobacillus* spp. that upregulate genes associated with lipid metabolism, has been shown to increase lipid absorption. These findings are supported by studies showing that GF mice exhibit upregulated cholesterol biosynthesis genes, increased lipid excretion and insulin sensitivity in response to high fat diet feeding, all of which contribute to altered lipid metabolism seen in these mice [26]. Multiple animal and human studies have identified the relationship between microbial lipids, systemic lipids and disease states and that various bacteria species are involved in lipid metabolism. However, determination of a direct causal relationship in humans poses several challenges due to interindividual variations in the composition of the microbiome, as complete microbiota depletion is not feasible. Notwithstanding, the murine studies provide strong causal support for the role of gut microbial communities and their metabolites in host lipid homeostasis (for review, see also [27]). 

Most human studies have examined generalized trends in microbiota composition as it pertains to dyslipidemia onset. More specifically, the relationship between levels of predominant bacterial species of the gut microbiota is often simplified to the ratio of Firmicutes to Bacteroidetes (F/B ratio) within an individual [28]. Changes in this F/B ratio are important in homeostatic processes and may contribute to various extraintestinal disease processes, such as dyslipidemias [11]. However, it is important to note that this ratio does not directly account for the less dominant species. More specifically, gut sequencing studies have associated certain microbial genera and metabolomic profiles to the development of dyslipidemia [29]. For example, adolescents with dyslipidemia exhibit diminished abundance of SCFA-producing genera *Akkermansia*, *Bacteroides*, *Roseburia*, and *Faecalibacterium* [30]. SCFAs have been demonstrated to increase Apolipoprotein A5 (Apo-A5), which is known to enhance uptake of TG, leading to an overall decrease in plasma TG [31]. Similarly, study findings have shown that adults with dyslipidemia harbor elevated concentrations of Gram-negative bacteria, such as *Escherichia coli* and *Enterobacter*, while decreasing beneficial bacterial species including *Lactobacillus*, *Faecalibacterium* and *Roseburia* [32]. Further, recent studies have also identified lower abundances of *Akkermansia*, *Roseburia* and *Bifidobacterium* in hypercholesterolemic individuals, and targeted microbiota interventions restored these deficits. Taken together, there is strong evidence demonstrating the link between relative deficiency of key bacteria species and altered lipid metabolism. Finally, metabolites produced by these gut microbiota species have been shown to modulate pathways associated with lipid synthesis and metabolism. In the following subsections, we further describe the roles that these metabolites play in dyslipidemia, particularly SCFA, trimethylamine *N*-oxide (TMAO), primary bile acids and coprostanol.

### 2.1. Short Chain Fatty Acids and Dyslipidemia

Short chain fatty acids (SCFA) are products of microbial metabolism of indigestible carbohydrates involved in multiple metabolic processes such as lipid synthesis, storage of fats, glucose uptake and inflammatory pathways [33]. They consist mainly of acetate, propionate, and butyrate. Different intestinal microbes produce different amounts of SCFAs. For example, acetate and propionate are mainly produced by Bacteroidetes while butyrate is produced mainly by Firmicutes [34,35]. Once synthesized, SCFA are absorbed through colonocytes and used as substrates for lipid, cholesterol, sugar or cytokine production [36,37,38]. SCFAs have long been shown to increase satiety and decrease food intake through activation of G-protein-coupled receptors [39,40,41] and regulate immune responses [42]. Given the wide range of roles SCFA play, their implication in dyslipidemia and atherosclerosis is not surprising. Dyslipidemia is characterized by a chronic low-grade inflammation and metabolic endotoxemia induced by dysbiosis changes that increase Gram-negative bacteria enteric species, an unfavorable state that can be improved by SCFA [43]. For example, SCFA have been shown to inhibit formation of foam cells and reduce production of proinflammatory cytokines [44]. More specifically, acetate is believed to affect IL-6 and IL-8 production through GPR41/43 activation, while butyrate and propionate affect IL-6 production [45]. In addition, they increase production of Treg cells and suppress histone deacetylase (HDACs) resulting in suppression of inflammatory response and therefore atherosclerosis development [46,47]. Further, the peroxisome proliferator-activated receptor gamma (PPAR-δ), a key regulator of lipid metabolism derived from the breakdown of fatty acids via beta-oxidation, has been shown to be dependent on dietary SCFAs [48,49]. As such, dietary SCFA are shown to influence PPAR-δ dependent pathways to reduce lipogenesis and promote beta-oxidation [50] as measured through increased mitochondrial AMPK activity (Figure 1A). AMPK is a highly conserved nutrient sensor that promotes lipid catabolism when AMP/ATP ratios become elevated in mitochondria [51]. More specifically, the SCFA, butyrate, is shown to promote adipogenesis through activation of the PPAR pathway resulting in reduced inflammatory and oxidative molecule [52,53]. Taken together, these findings support the role of SCFA in lipid metabolism through PPARδ activation.

Among bacteria, the *Enterobacteriaceae* family of bacterial species, that includes genera *Escherichia coli* and *Enterobacter* were found to be elevated in dyslipidemia patients, which was associated with production of cytotoxic amounts of nitric oxide [54]. Excess nitric oxide and other oxidative species induce formation of oxidized LDL, triggering inflammation and plaque formation in the vasculature [55], resulting in the onset of clinical sequelae of dyslipidemia like coronary artery disease. Importantly, PPAR-δ also serves as an inhibitor of inducible nitrate oxide synthase genes [56]. In this regard, butyrate was found to reduce endothelial NADPH oxidase 2 (Nox2) and reactive oxygen species by upregulating PPAR-δ/miR-181b pathway preventing endothelial dysfunction seen in atherosclerosis [45,57]. Therefore, SCFA-producing genera, characteristically reduced in dyslipidemia patients, including *Akkermansia*, *Bacteroides*, *Roseburia*, and *Faecalibacterium* play a critical role in maintaining lipid balance and counteracting unfavorable effects of pathogenic *Enterobacteriaceae*.

Butyrate also can reduce HMG-COA reductase gene expression resulting in reduction of cholesterol biosynthesis [58]. Several studies have shown that butyrate supplementation in high fat fed mice caused reduction in low density lipoprotein cholesterol and total cholesterol [59]. Similarly, propionate has been shown to decrease cholesterol serum levels by inhibiting acetate incorporation into fatty acids and cholesterol [34]. For example, in ApoE^–/–^ mice, treatment with propionate was associated with reduced atherosclerotic lesion burden [60]. Similar findings have been seen in humans. In a randomized, double-blinded, placebo-controlled human study, administration of 500 mg of propionate twice a day was associated with significantly reduced LDL and non-high density lipoprotein cholesterol levels compared to the control group [61]. Taken together, these studies show that administration of SCFA such as butyrate and propionate may serve as a treatment modality for atherosclerosis. Further research, however, needs to be conducted to determine the specific role played by these SCFA.

### 2.2. Trimethylamine N-Oxide (TMAO)

Trimethylamine *N*-oxide (TMAO) is a gut metabolite produced from the oxidation of trimethylamine, an intermediate compound, by hepatic flavin monooxygenase (FMO1 and FMO3) [62]. The gut microbiota plays a critical role in converting dietary choline into trimethylamine (TMA) which ultimately gets oxidized to TMAO in the liver [63]. Diet rich in dairy products, eggs, red meats, fish and other seafood are potential sources of TMAO [64,65]. TMAO came into the spotlight when it was discovered that elevated levels of TMAO were associated with atherosclerosis and risk of CVD. In a study involving 220 subjects, increased TMAO was associated with increased carotid intima media thickness, an early marker for atherosclerosis [66]. Additional research involving patients presenting to emergency department for cardiac related chest pain found elevated TMAOs levels to be associated with risk of major adverse cardiac events suggesting its role as a biomarker [67]. Nine bacteria strains have been shown capable of producing TMA from choline suggesting that alterations in the gut microbiota can have marked effects on TMAO levels [68]. In mice, transplantation of choline converting bacteria to germ free mice has been shown to increase TMA production [69,70]. Furthermore, use of broad-spectrum antibiotics causing changed in the gut microbiota resulted in near suppression of TMAO levels [71]. The use of vancomycin, metronidazole vancomycin, neomycin-sulphate, metronidazole, and ampicillin have all been shown to inhibit choline enhanced atherosclerosis [72]. Likewise, increases in TMAO levels after phosphatidylcholine challenge were markedly reduced following administration of antibiotics in healthy participants providing possible therapeutic avenue for atherosclerosis development [71]. Taken together, these studies suggest that increased levels of TMAO may result from dysbiosis of gut microbiota which may in turn further the pathogenesis and progression of dyslipidemia. 

Although many studies have demonstrated the association of TMAO and development of atherosclerotic cardiovascular disease, its proposed mechanism remains unclear. TMAO involvement is multifactorial with the summation of effects contributing to hyperlipidemia and cardiovascular disease [69,73]. First, TMAO has been shown to upregulate scavenger A and CD36 located on macrophages with a role in the uptake of oxidized LDL via MAPK/JNK pathway [74] (Figure 1B). This leads to formation of foam cells, a hallmark of atherosclerotic disease. In mice, where production of TMAO was inhibited by antibiotics, the number of macrophages and formation of foam cells were drastically reduced [72]. The association of TMAO and atherosclerosis, however, is not all that clear. In a study involving APoE^−/−^ mice expressing human cholesteryl ester transfer protein (hCETP), TMAO was inversely correlated with aortic lesion size suggesting a possible protective role in aortic lesion formation [75]. As far as lipid metabolism, TMAO has been shown to play a role in decreased reverse cholesterol transport (RCT) [76]. RCT is a multi-step process involving removal of excess cholesterol from peripheral tissue and delivering it to the liver for redistribution or removal through bile acid [77]. Further disruption of bile acid regulation is seen as TMAO has been shown to be negatively related to bile acid pool size through inhibition of bile acid synthesis and liver bile acid transportation. Farnesoid X receptor (FXR) is a nuclear receptor family activated by bile acids regulating bile acid synthesis and transport. TMAO downregulates CYP7A1 and CYP27A1 enzymes involved in the bile acid synthesis through activation of FXR [78]. In Ldlr-/- mice, studies have shown that downregulation of FXR results in decrease TMAO levels and reduction in the size of atherosclerotic lesions in the aorta. Interestingly, this result was found in male but not female mice [79]. In another study, Bennett et al. found that TMAO levels explained about 11% of difference in atherosclerosis susceptibility in female mice suggesting a possible role of TMAO in atherosclerosis development [80]. Moreover, expression of Niemann-Pick C1-like1 (Npc1L1), cholesterol transporter from intestinal lumen into enterocytes, and ABCG5/8, which transports cholesterol out of enterocytes into the gut lumen, is significantly decreased with TMAO dietary supplementation [81]. This may be an additional mechanism through which TMAO affects atherosclerosis. Therefore, these cumulative effects of TMAO make for an interesting treatment opportunity for CVD reduction.

### 2.3. Primary Bile Acids

Bile acids are hydroxylated steroids that act as emulsifiers, aiding in the process of solubilization and digestion of dietary lipids. They serve as the primary pathway for breakdown and excretion of cholesterol playing a role in cholesterol, triglyceride and energy regulation [82]. Synthesis of bile acid (primary bile acid) occurs in the liver and involves hydroxylation of cholesterol by rate limiting enzyme cholesterol 7α-hydroxylase (CYP7A1) [83]. Two primary bile acids, cholic acid and chenodeoxycholic acid, are the end result of the bile acid synthetic pathway. These products are typically conjugated with glycine in humans, excreted into bile and facilitate emulsion following consumption of a meal [84]. Roughly 95% of the bile acids are reabsorbed from the intestine at the distal ileum to be transported back to the liver via the portal circulation [85]. Primary bile acids can also be acted upon by intestinal bacterial flora to form secondary bile acids: deoxycholic and lithocholic acid, derived from cholic acid and chenodeoxycholic acid, respectively [86]. When the bile acid pool increases, bile acid binds to nuclear receptor farnesoid X receptor (FXR). FXR induces production of inhibitory nuclear receptor SHP that interacts with nuclear receptor living receptor homolog (LRH)-1 suppressing gene transcription of the CYP7A1 gene resulting in decreased synthesis of primary bile acid [83,87] (Figure 1C). Furthermore, FXR promotes production of fibroblast growth factor 15/19 in intestine resulting in bile acid synthesis inhibition in the liver and increased cholesterol into intestinal lumen through ABCG5/ABCG6 cholesterol exporter [88]. Bile acids can also act on membrane Takeda G-protein-coupled receptor 5 (TGR5). Activation of TGR5 has been shown to promote energy expenditure [89,90]. Disruption of this receptor has been shown to decrease total bile acid pool size and increase fat accumulation with body weight gain in female mice [91]. In obese mice, TGR5 has been shown to improve liver and pancreatic function by inducing GLP-1 release from enteroendocrine L-cells, resulting in improved liver and pancreatic function in obese mice [92]. Current literature suggests some interplay between the two. Bile acids can also alter the structure of the gut microbiota. Bile acids produce an environment that favors bacteria resistant to BA, promoting microbial diversity [93]. Conjugated bile acids have been shown to prevent bacterial overgrowth and promote epithelial cell integrity through binding to FXR and inducing antimicrobial proteins angiogenin 1 and RNase family member 4 [94]. In a study involving ApoE-/-, a polyphenol, resveratrol, altered gut microbiota composition by increasing levels of *Bifidobacterium* and *Lactobacillus* which increased levels of deconjugated BA. This, in turn, altered and reduced atherosclerosis progression [95]. It has also been shown that specific species like *Lactobacillus reuteri*, decrease LDL cholesterol while increasing bile acids through their bile salt hydrolase activity [96], which deconjugates bile acids.

Further research needs to be conducted to better understand the role bile acids play in dyslipidemia as activation and inhibition of FXR by BA show contradictory findings. Individuals with obesity have been shown to have increased BA synthesis, preferential 12α-hydroxylation and decreased serum BA fluctuations [97,98]. 12α-hydroxylated bile acids are more effective in emulsifying dietary fat suggesting that obesity may alter the composition of bile acids that in turn promotes an environment for dyslipidemia [98]. In high-fat diet-induced obese (DIO) mice, reduction in 12α-hydroxylated bile acids through bile acid synthesis enzyme CYP8B1 improved oral glucose tolerance, and reduced liver triglycerides helping managing dyslipidemia [99,100]. Recent research has also shown insulin suppression of FoxO1, an insulin receptor, results in reduced 12α-hydroxylated bile acids, cholesterol absorption, and plasma cholesterol levels serving as a possible mechanism for prevention of hypercholesteremia. Restoring insulin signaling or reducing 12α-hydroxylated bile acid levels normalized cholesterol levels suggesting the possibility that targeting cholesterol absorption rather than synthesis may prove to be more beneficial particularly in those with type 1 diabetes [101]. Taken together these studies show that, disruption of the bile acid pathway is seen in individuals with metabolic disease, thus targeting this pathway may serve as a potential therapeutic option.

### 2.4. Microbial Cholesterol Dehydrogenases

Microbial enzymes have been shown to have the capacity to convert cholesterol to non-absorbable metabolites that can be excreted, thus reducing serum cholesterol levels [99]. For example, the microbial gene, intestinal sterol metabolism A (i*smA*), is responsible for encoding a family of cholesterol dehydrogenase enzymes that convert circulating cholesterol to coprostanol [102] (Figure 1D). Coprostanol is a form of cholesterol that is eliminated through feces. Although the role of coprostanol is not fully understood at this time, individuals with microbiota expressing i*smA* have been shown to have lower total serum cholesterol [102]. Certain bacteria have been associated with coprostanol-forming capacity that have also been shown to be decreased in dyslipidemia. These bacteria include *Lactobacillus* spp., *Bifidobacterium* spp. and *Eubacterium* [102,103]. Phylogenetic evaluation and gut sequencing studies also showed that *Faecalibacterium prausnitzii* and *Clostridium leptum* encode *ismA*, therefore having the ability to metabolite cholesterol into coprostanol. As such, shifting the gut microbial composition to increase abundances of these genera and species can serve to improve dyslipidemia through formation of coprostanol, a gut microbiota metabolite derived from cholesterol. Still, further research is needed to fully elucidate the mechanisms and specific bacteria expressing i*smA* and have coprostanol forming capacity. Additionally, other microbial enzymes such as bacterial stereospecific hydroxysteroid dehydrogenases (HSDH), bile salt hydrolases (BSH) and bile acid-inducible (BAI) enzymes can indirectly have favorable roles on dyslipidemia through modulation of bile acids [104]. HSDH enzymes modify bile acids by way of oxidation, epimerization and dehydroxylation, whereas BSH and BAI enzymes perform bile acid deconjugation dehydroxylation, therefore promoting bile acid homeostasis and changes in lipid parameters [105]. Numerous Gram-positive and Gram-negative bacteria can produce these enzymes, including *Lactobacillus*, *Bifidobacterium* and *Bacteroides.* Further, BSH enzymes provide a dual benefit as they have been shown to confer protection for certain bacterial species in the GI tract through deconjugation of toxic bile acids along with its role in modification of bile acids to forms that are readily excreted. Taken together, these findings support the role of inherent microbial enzymes that can either directly conjugate cholesterol to a form that is more easily excreted via *ismA* or indirectly through bile acid modulation via enzymes such as HSDH, BSH and BAI.

## 3. Diet and Its Effects on Gut Microbiota and Dyslipidemia

Diet is understood to be the most important determinant in shaping the microbiota ecosystem as balanced diets of fruits and vegetables are shown to increase gut bacterial richness and diversity [10,15]. As such, lifestyle changes specifically through dietary interventions represent an effective therapeutic modality for dyslipidemia by creating targeted and beneficial changes in gut microbial composition. Although many dietary choices influence the composition of the gut microbiota, two particular diets have been studied extensively and can lead to significant, yet opposite effects. In this section, we first describe the metabolic changes seen in Western Diet (WD)-induced dyslipidemia followed by the role of plant-based diets such as the Mediterranean diet (MD) play in optimizing gut microbial composition to restore defective metabolic states.

### 3.1. High Fat Diet, Gut Microbiota and Dyslipidemia

High fat diets (HFD), such as the WD, are comprised by increased amounts of animal proteins, saturated fats and sugars, with low levels of fibers and phenols. Additionally, chronic HFD intake results in decreased total microbiota richness with a shift towards overgrowths of lipopolysaccharides (LPS) containing, Gram-negative enteric bacteria, which are shown to contribute to metabolic endotoxemia [106,107]. More specifically, elevated concentrations of circulating LPS induced by a chronic HFD facilitates increased binding to its receptor, Toll-like receptor 4 (TLR4), to activate innate and adaptive immunity triggering a pro-inflammatory cascade [108]. TLR4 activation has been implicated in the development of dyslipidemia and associated atherosclerosis, through enhanced release of pro-inflammatory cytokines like IL-8, IL-1beta and TNF-alpha [109]. Results from this study also showed TLR-4 mediated oxidation of low-density lipoproteins, which are known to accumulate in unstable plaques that deposit in the vasculature. As such, the combination of altered lipid metabolism as seen through increased LDL combined with LPS/TLR-4 mediated oxidization, worsens dyslipidemia leading to more severe associated clinical sequelae (Figure 2). It is also shown that lipoproteins, particularly HDL, has a protective effect against LPS-mediated metabolic endotoxemia by facilitating clearing of LPS from circulation [108]. Therefore, decreases in HDL and increases in relative concentrations of unfavorable Gram-negative bacteria resulting from HFD-induced dyslipidemia, further worsens hypercholesterolemic states through inability to clear excess LPS. 

Further, it has been shown that microbial species like *Escherichia coli* and *Ruminococcus* spp., that are elevated after HFD feeding, contribute to increased intestinal barrier permeability, allowing for LPS to translocate into the blood stream and cause systemic low-grade inflammation characteristic of dyslipidemia [110]. In turn, high-fat diets and associated dyslipidemia chronically activate the NLRP3 inflammasome, which plays a role in inducing macrophage activity and release of proinflammatory cytokines like interleukin-1 (IL-1) [111]. In particular, sterol regulatory element binding protein 2 (SREBP-2), a transcription factor that facilitates lipogenesis, complexed with SREBP cleavage-activating protein (SCAP) contributes to NLRP3 inflammasome assembly [112]. This, in turn, affects cholesterol biosynthesis signaling with NLRP3 induced inflammation in macrophages. Excessive NLRP3-mediated release of proinflammatory cytokines has been associated with downregulation or decreased expression of the LDL-receptor [113]. The LDL-receptor has been shown to be downregulated in chronic high fat intake [114]. Taken together, these findings show how chronic high-fat diets can contribute to metabolic endotoxemia and dyslipidemia through upregulation of key pro-inflammatory factors including NLRP3. 

### 3.2. Mediterranean Diet, Gut Microbiota and Dyslipidemia

The Mediterranean Diet (MD) consists of plant-based ingredients including wheats, cereals, nuts, fruits, vegetable, omega-3 polyunsaturated fatty acids, with high amounts of fibers and polyphenols [8]. Collectively, these ingredients have been shown to have beneficial effects on lipid imbalances through their antioxidant and anti-inflammatory effects [115,116] as shown via reduction in TNF-alpha and LPS concentrations [117,118]. Further, diets high in polyphenols, known as indigestible phytochemicals, that are abundant in plant-based foods, increase production of beneficial SCFAs because of their interaction with colonic microbiota [119]. For example, the polyphenol, resveratrol, specifically has been found to increase the SCFA-producing bacterial genera, *Allobaculum*, *Bacterioides*, and *Blautia* while also inhibiting TLR-4 [120,121,122]. Further, the rich plant-based MD increases fiber-derived SCFA by the gut microbiota, such as acetate, propionate, and butyrate [119] and has been shown to decrease total cholesterol and LDL cholesterol [123] (Figure 2). As described in the previous sections, SCFA exert a myriad of beneficial effects including attenuating the progression, or preventing against, the onset of dyslipidemia.

Several studies have shown that low carbohydrate (LC) diets significantly improve the Bacteroidetes/Firmicutes ratio, as well as important metabolic markers of dyslipidemia [124]. More specifically, the LC diet reduced fatty acids associated with de novo lipogenesis pathways, while increasing omega-3 PUFA that are shown to exert anti-inflammatory and anti-hypertriglyceridemia effects [125]. Importantly, microbiota changes after LC were also associated with increased HDL and decreased triglycerides. This is supported by findings showing that vegetable oils particularly omega-3 PUFA and phytonutrients, had beneficial effects on serum lipid profiles including HDL, TG, and apolipoprotein B in hypercholesterolemic patients [126]. Hypocholesterolemia has been associated with increased levels of *Clostridium leptum*, known to be involved in promoting increased cholesterol and bile acid metabolism [127]. Therefore, vegetable oils such as omega-3 PUFA, a key component of the MD has an important role in maintaining a healthy lipid balance. 

Further, oatmeal, a food option within the umbrella of the MD, improve cardiometabolic parameters in patients with dyslipidemia through key changes in gut microbiota profile [128]. These changes include increased beneficial bacteria like *Akkermansia* along with SCFA-producing *Faecalibacterum*, *Dialister*, *Lactobacillus*, with decreases within the *Rumminococcacae* family. *Akkermansia* spp. are shown to improve metabolic parameters through reducing oxidative stress, reducing fat mass, insulin resistance and dyslipidemia in a rodent model [129,130]. Therefore, these findings of elevated *Akkermansia* and SCFA-producing bacterial species after oatmeal were associated with decreased total cholesterol and LDL with increased amounts of serum antioxidant capacity. Further, Sun et al. showed that oat-based foods (OF) rich in beta-glucans, when compared to high-fat diets (HFD) and control diets had significant reductions in plasma total cholesterol (TC), low-density lipoprotein (LDL), and triglycerides (TG), along with increased concentrations in several SCFAs including butyrate, propionate, and acetate [131]. Similarly, it has recently been shown that flavonoids, a component of whole-grain oat, regulates bile acid pathways to reduce hyperlipidemia induced by chronic HFD feeding [132]. Flavonoids upregulate expression of PPAR, carnitine palmitoyl transferase I (CPT-1) and FXR, while down-regulate SREBP-1 and fatty acid synthase (FAS). As mentioned, PPAR contributes to the breakdown of fatty acids via beta-oxidation by inducing CPT-1 activity while FXR regulates bile acid synthesis and transport by promoting efflux to feces to improve dyslipidemia. On the other hand, SREBP-1 induces lipogenesis in the liver and promotes fat storage in the form of triglycerides [133], therefore down-regulating this transcription factor is beneficial in states of dyslipidemia. Additionally, *Akkermansia* is increased in flavonoid treated mice, while unfavorable species associated with a HFD like *Desulfovibrio* was decreased. As such, *Akkermansia* can serve an important role in improving dyslipidemia not only through dietary interventions, but as a potential next-generation probiotic, discussed in a future subsection. Taken together, these findings provide strong evidence for MD as a lifestyle intervention that contributes to generalized and specific favorable changes in the gut microbiota composition to improve metabolic parameters.

### 3.3. Current Pharmacologic Treatments for Dyslipidemia and Relations to Gut Microbiota 

Pharmacologic interventions of dyslipidemia may include inhibition of cholesterol synthesis, increasing use of cholesterol for bile acid production, or conversion of cholesterol in the gut to non-absorbable metabolites. Interestingly, the current pharmacological treatment modalities are associated with changes in gut microbiota, indicating the potential for synergistic effects of medications and targeted gut microbiota therapy for dyslipidemia.

Statin drugs are the first-line agents to reduce cholesterol synthesis by inhibition of HMG CoA Reductase. Use of statins has been demonstrated to improve the composition and function of gut microbiota [134] and has been associated with lower occurrence of gut microbiota dysbiosis [135]. For example, statin responsive patients showed increased concentrations of SCFA-producing genera including *Lactobacillus*, *Eubacterium*, *Faecalibacterium* and *Bifidobacterium*, all of which are characteristically decreased in dyslipidemia patients [136]. On the other hand, statin resistant patients did not exhibit similar changes. Further, other studies support the changes that statins can exert on gut microbiota, particularly elevations in *Blautia* and *Bifidobacterium longum*, which has specifically been correlated with decreased triglycerides and overall body weight [134]. As such, these combined findings suggest that response to statin treatment may, to some extent, be mediated or predicted by alterations in gut microbiota

Bile acid sequestering agents in the treatment of dyslipidemia are used to partially remove bile acids from enterohepatic cycling to increase use of cholesterol in the synthesis of new bile acids [137]. As mentioned earlier, bile acids have been shown to have favorable effects on gut microbiota. Therefore, it is possible that bile sequestering agents such as Cholestyramine, can mediate beneficial effects on dyslipidemia through changes in gut microbiota. For example, HFD and cholecystectomy mice treated with cholestyramine exhibit improved lipid profiles, beneficial effects on PPARδ and SREBP1 concentrations while concurrently elevating concentration of favorable gut bacteria including *Blautia*, *Alistipes* and *Eubacterium* [138]. It has also recently been shown that cholestyramine increased SCFA and enriched concentrations of SCFA-producing *Lachnospiraceae* spp. in treatment responsive groups [139]. More generally, cholestyramine increased the Bacteroidetes/Firmicutes ratio, indicating a shift towards a more favorable gut composition profile with overall reductions of inflammatory markers. These anti-inflammatory effects are supported by results showing that cholestyramine can reduce inflammatory signaling in HFD-induced mice [140]. Therefore, bile acid sequestrants are able to influence gut microbiota via bile acid-microbiota crosstalk to improve dyslipidemia, though more studies are needed to elucidate the specific mechanisms by which they do so. 

Although not first-line for dyslipidemia treatment, metformin can also improve dyslipidemia in patients with T2DM through mechanisms that increase insulin sensitivity to reducing LDL and TG [141]. Recent findings have shown that metformin exerts favorable changes in gut microbiota, most notably through increases in *Blautia* and *Faecalibacterium*, which were associated with lipid homeostasis and improvements in serum triglyceride levels [142]. Further, metformin has been shown to increase concentrations of *Akkermansia* while improving total cholesterol levels in a rodent model of metabolic syndrome [143]. Therefore, these changes in gut microbiota can provide more insight into mechanisms by which pharmacological therapy can augment microbiota mediated pathways to exert their effects.

## 4. Targeted Microbiota Therapies

In recent years, there has been increasing research describing the role of targeted microbiota therapy in improving lipid metabolism. Generally, the goal of targeted microbiota therapy is to create favorable changes in gut microbial composition, to increase its richness and diversity, specifically by enhancing richness of particular genera that are influential in treating dyslipidemia. However, it is important to keep in mind that diet, lifestyle, and other environmental changes may concurrently affect the structure, composition and metabolic function of gut microbiota and often must be considered along with these therapeutic interventions. Overall, these treatment modalities include prebiotics, probiotics, synbiotics, folate and fecal microbiota transplants. In the following subsections, we describe these treatment methods, how they improve gut microbial composition profile and the mechanisms by which they may exert their effects in dyslipidemia.

### 4.1. Prebiotics

Prebiotics (Table 1) are organic compounds that can be utilized by the symbiotic gut microbiota to support their growth. By providing an energy source for commensal bacteria, prebiotics are shown to positively impact humans’ health via modulation of lipid metabolism, improving intestinal barrier functioning, intestinal cell growth, proliferation, and absorption of minerals such as iron, magnesium, and calcium [144]. Compounds that are prebiotic in nature can be found in a wide variety of foods such as whole wheat bread, fruits, vegetables, herbal teas and even orange juice [145], though extracts have been used to promote targeted changes in gut microbiota. These compounds include beta-glucans, psyllium, and inulin, which will be further discussed. 

Beta-glucans, used in prebiotic supplements, are a group of non-starch polysaccharides found in oats and wheats and have demonstrated many health benefits, from modulating gut microbial concentrations to pro-immune effects and improving serum cholesterol levels. Beta-glucans support the growth of beneficial species like *Bifidobacteria*, *Akkermansia* and *Lactobacilli* in vivo as well as in vitro [146,147]. Further, beta-glucan contained in oat fibers improves lipid profiles, particularly reducing serum LDL by 15% and total cholesterol by 8.9% in an 8-week trial period [148]. A recent study showed that beta-glucan consumption for as little as 4 weeks reduced LDL cholesterol by 6% and overall cardiovascular disease risk by 8% [149]. Similarly, ingestion of 3 g of soluble fibers from oats for one-month, significantly reduced both total cholesterol and LDL cholesterol compared to a control group [150]. It is also important to note that oats comprise a portion of the MD and beta-glucans can be obtained from diet adherence as well as in prebiotic supplements. Aside from gut microbiota modulation, beta-glucans possess host immunomodulatory effects that can also improve dyslipidemia. This is evidenced through studies that show that beta-glucans reduced pro-inflammatory cytokines and overall inflammation in diet-induced dyslipidemia [151]. Further, beta-glucans are shown to interact with innate immune cells via receptors such as dectin-1 and complement receptor type 3 (CR3) which have been associated with several immunoregulatory processes [152]. These changes can be attributed to beta-glucan mediated increases in SCFAs through restructuring the gut microbial composition [153]. As such, these immunomodulatory effects serve to improve hypercholesteremic states as well as prevent the progression to atherosclerosis and other more severe sequalae. Lastly, beta-glucans positively influence cholesterol and bile salt regulation. These compounds seem to exert their lipid-lowering abilities due to an inherent ability to increase the viscosity of fecal matter [154] that can prevent reabsorption of bile salts. As a result, the liver must increase production of new bile salts which involves increased uptake of circulating cholesterol, leading to a reduction in serum cholesterol levels [147]. Overall, beta-glucans have favorable and targeted effects on gut microbiota that can improve lipid profiles in dyslipidemia patients. 

Further, psyllium, a viscous dietary fiber, used as a prebiotic is shown to including improvements in lipid balance through targeted changes in gut microbiota. A recent meta-analysis involving 28 trials found that supplementation of a median dose of 10.2 g of psyllium yielded significant reductions in LDL, non-HDL cholesterol, and apolipoprotein B [155]. It was suggested that alterations in viscosity by psyllium, increases utilization of serum cholesterol by the liver in de novo bile acid synthesis, a mechanism similar to how beta-glucans may exert their benefits. Additionally, psyllium significantly increase concentrations of butyrogenic species including *Roseburia*, *Lachnospira* and *Faecalibacterium* concentrations and associated SCFAs, while also improving immune function [156]. Interestingly, recent findings suggest that psyllium husk is more effective than orlistat, a lipase inhibitor, in reducing liver cholesterol and TG levels in HFD-induced obesity [157]. Both psyllium husk and orlistat had beneficial effects on FXR receptor and sterol-27-hydroxylase expression, while regulating levels of important bile acids in the feces. It should also be noted that this study supports increases in *Faecalibacterium*, *Roseburia* and *Akkermansia* reported in other studies assessing psyllium intervention, further indicating the important of these bacterial species in lipid balance.

Inulin is another naturally occurring polysaccharide, commonly used in prebiotic supplements and found to improve weight loss, diabetes and lipid imbalance [158]. In combination with 2 g of phytosterols, 10 g inulin-enriched soymilk decreased LDL cholesterol by 9% and total cholesterol by 5% more than standard soymilk. Inulin prebiotics induce characteristic changes in gut microbiota including increase in *Bifidobacteirum*, *Faecalibacterium* and *Lactobacillus* and decreased *Bilophila* [159,160] with associated increases in SCFA. *Bilophila* is a sulfate-reducing bacteria that yields byproducts including hydrogen sulfide, which is shown to inhibit butyrate [161]. Further, a combination of increased *Bifidobacterium* and decreased *Bilophila* is associated with decreased pro-inflammatory cytokine. including IL-1, TNF-alpha and IL-10 [162]. As such, this favorable alteration of gut microbiota contribute to improving lipid profiles by decreasing inflammation and increasing butyrate concentrations and activity.

**Table 1 nutrients-15-00228-t001:** Targeted Microbiota therapy studies, outcomes, results, and implications.

Targeted Microbiota Therapy Method	Study Period	Species Involved/Outcome Measured	Results/Implications	Subject Type	Reference
Prebiotic—Beta-glucans		*Bifidobacterium*, *Lactobacillus*	Increased SCFA production Decreased cholesterol biosynthesis	Mice	[147]
Prebiotic—Beta-glucans (oat and tartary buckwheat)		Bacteroidetes/Firmicutes ratio	Increased SCFA production Reduction of plasma lipids Increased fecal bile acid concentration	Rodent	[131]
Prebiotic—Beta-glucans			Strong immunomodulary effects Reduced serum cholesterol levels		[152]
Prebiotic—Oatmeal	45-day follow-up	*Akkermansia*, *Dialister*, *Faecalibacterium*, *Barnesiella*, *Agathobacter*, *Lactobacillus Ruminococcaceae*-MK4A214	Increased *Akkermansia*, *Dialister*, *Faecalibacterium*, *Barnesiella*, *Agathobacter*, *Lactobacillus* Decreased *Ruminococcaceae*-MK4A214 Decreased serum TC, LDL, and non-HDL cholesterol Increased serum total antioxidant capacity Increased SCFA production	Human	[128]
Flavonoids from whole-grain oat		*Akkermansia*, *Blautia Lachnoclostridium*, *Colidextribacter*, *and Desulfovibrio*	Improved serum lipid profiles Decreased body weight Decreased lipid deposition Increased *Akkermansia* Decreased *Lachnoclostridium*, *Blautia*, *Colidextribacter*, *and Desulfovibrio*	Mice	[132]
Prebiotic—Wheat bread and barley beta glucans	4 weeks	*Akkermansia muciniphila* & *Bifidobacterium* were elevated pre-intervention in cholesterol-responsive group	Decreased abdominal circumference Decreased total cholesterol Increased fecal propionic acid Decreased *Clostridium leptum* by 25% and *Collinsella aerofaciens*, a species that thrives within inflamed gut tissues	Human	[146]
Prebiotic—Oat beta-glucans	8 weeks	Serum lipids	Reduced LDL, TC, and non-HDL in mildly hypercholesterolemic patients	Human	[148]
	4 weeks	Serum lipids	Reduced LDL by 6% 8% reduction in CVD risk	Human	[149]
	4 weeks		Reduced serum TC and LDL	Human	[150]
			Lowered markers of inflammation in heart/liver/kidney/spleen/colon in obese mice fed high-cholesterol diets	Mice	[151]
	30 days	Acetic acid Propionic acid Hydroxybutyric acid	Reduction in mucosal damage—Increased fecal concentrations of acetic acid, propionic acid, and hydroxybutyric acid Decreased serum CRP	Human	[153]
Prebiotic—Psyllium (plantago ovata) fiber	Meta analysis of 28 trials greater than or equal to 3 weeks	N/A	Significant reduction in LDL cholesterol, non-HDL cholesterol, and apoB lipoproteins	Human	[155]
Prebiotic—Psyllium husk	7 days	*Roseburia*, *Lachnospira*, and *Faecalibacterium*	Increased concentrations of *Lachnospira*, *Faecalibacterium*, *Phascolartobaceterium*, *Veillonella*, and *Sutterella* Increased fecal water content associated with increased butyrate-producing strains (*Lachnospira*, *Roseburia*, and *Faecalibacterium*)	Human	[156]
		*Roseburia Bacteroides*, *Faecalibacterium*, *Coprobacillus*, *and Akkernansia*	Greater reduction in cholesterol and TGs compared to Orlistat	Mice	[157]
Prebiotic- Inulin-type fructans		*Bifidobacterium*, *Faecalibacterium*, *Lactobacillus*	Increased insulin sensitivity Increased gut barrier function Improved lipid profiles		[158]
		*Bifidobacterium*, *Anaerostipes*, *Bilophila*	Increased *Bifidobacterium* and *Anaerostipes* Decreased *Bilophila*	Human	[159]
	6 weeks	*Bifidobacterium* *Acetic acid*, *propionic acid*, *butyric acid*	Significantly increased *Bifidobacterium* Increased total fecal SCFA, acetic acid and propionic acid in Type 2 DM patients	Human	[160]
Dietary glycan—Seaweed Polysaccharide	6 weeks and 12 weeks	*Bifidobacteria*, *Akkermansia*, *Pseudobutyrivibrio*, *Clostridium*, *Bilophila*	Significantly reduced non-HDL cholesterol Increased *Bifidobacteria*, *Akkermansia*, *Pseudobutyrivibrio* and *Clostridium* Decreased *Bilophila*		[162]
Probiotic- *Lactobacillus*, *Bifidobacterium*, *Streptococcus*	6 weeks	*Lactobacillus*, *Bifidobacterium and Streptococcus*	Decreased fasting plasma glucose versus control group Increased serum HDL versus control group	Human	[163]
Probiotic—*Lactic acid producing strains*		*Lactobacillus casei*, *Lactobacillus paracasei*, *Lactobacillus plantarum*, *Enterococcus faecium*, *Enterococcus lactis*	Incorporation of probiotics into foods containing dairy reduced reduced serum cholesterol		[164]
Probiotic—*Bifidobacterium bifidum*	3 weeks	*Firmicutes*, *Bacteroides*, *Actinobacteria*, *Proteobacteria*, *Fusobacteria*, *Dorea*, *Lachnospira*	Increased *Firmicutes*, *Bacteroides* and *Actinobacteria* Decreased in *Firmicutes*, *Bacteroides* and *Actinobacteria* Decreased in total cholesterol and LDL cholesterol	Human	[165]
Probiotic milk—*Lactobacillus acidophilus*, *Lactobacillus casei*, *Bifidobacterium lactis*	10 weeks supplement plus 2 weeks follow-up	*Lactobacillus acidophilus*, *Lactobacillus casei*, *Bifidobacterium lactis*	Improved fecal weight Decreased fecal passing time Increased biodiversity of *Lactobacillus* and *Bifidobacterium* spp. Improved lag-time of LDL oxidation Decreased serum cholesterol	Human	[166]
Probiotic—*Bifidobacterium animalis* subsp. *lactis*	6 months	*Lactobacillus* and *Akkermansia*	Significantly increased fecal *Bifidobacterium*, *Akkermansia*, *and Streptococcus* in supplemented group Decreased glycocholic acid, glycoursodeoxycholic acid, taurohyodeoxycholic acid, and tauroursodeoxycholic acid	Human	[167]
Synbiotic—xylo-oligosaccharides (XOS) + *Bifidobacterium animalis lactis*	3 weeks	XOS + *Bifidobacterium animalis lactis*	Increased host Th1 responses, increase in HDL, increased *Bifidobacterium* count	Human	[168]
Synbiotic—xylo-oligosaccharides (XOS) + *Bacillus licheniformis*		XOS + *Bacillus licheniformis*	Reduction in serum LPS, decreased body weight, decreased serum total cholesterol	Mice	[169]
Folate			Reduced body weight gain, adipocyte size and dysbiosis Down-regulated lipid-metabolism genes	Mice	[170]
			Lower serum folate levels were associated with increased carotid intima-media thickness	Human	[171]
		*Porphyromonadaceae*	Low folate diet resulted in higher amounts of *Porphyromonadaceae* and associated NAFLD	Mice	[172]
Fecal Microbiota Transplant		*Bifidobacterium*, *Lactobacillus*, *Bilophila and Desulfovibrio*	Increases in *Bifidobacterium* and *Lactobacillus* Decreased *Bilophila* and *Desulfovibrio*	Human	[173]
	24 weeks	*Bifidobacterium* and *Lactobacillus*	Increases in butyrate-producing bacteria Improvements in total cholesterol and LDL		[174]
	12 weeks	Fecal bacteria Bile acids	Decreased taurocholic acid versus baseline Bile acid profile shifts towards that of the donor		[175]
*Akkermansia muciniphila*		*Akkermansia muciniphila*	Significant positive correlation with PUFA/SFA ratio Negatively correlated with onset of dyslipidemia Reduced body fat mass and insulin resistance Increased tight junction proteins, zonulin-1 and occludin Increased IL-10 Degradation of host mucin lining	Human	[176]
		*Akkermansia muciniphila*	Improved gut barrier function via interactions with TLR-2	Mice	[129]
		*Akkermansia muciniphila* and *Periplaneta americana extract* (PAE)	PAE pretreatment greatly increased amount of *Akkermansia muciniphila* versus control facing diquat-induced oxidative stress	Mice	[130]
		*Akkermansia mucinophila*	Increased therapeutic effect of the novel anti-hyperlipidemic plant-alkaloid, Nuciferine, via enrichment with *Akkermansia mucinophila*	Mice	[177]
		*Akkermansia mucinophila*	Increased Akkermansia muciniphila was associated with decreased risk of metabolic syndrome once A. muciniphila comprised 0.2% of total microbiome	Human	[178]
*Faecalibacterium prausnitzii*		*Faecalibacterium prausnitzii*	Mononuclear cell stimulation of *Faecalibacterium prausnitzii* lowered IL-12 and IFN-gamma production Increased secretion of IL-10 Displayed anti-inflammatory effects including blocking NF-KB and IL-8 production		[179]
		*Faecalibacterium prausnitzii*	Produced butyrate thereby inhibiting NF-KB, and downregulating TLR-3/TLR-4 Stimulated mucin secretion, improving gut barrier functionality		[180]
		*Faecalibacterium prausnitzii*	Decreased abundance of the species in obese individuals Exhibited anti-inflammatory effects Produced butyrate		[181]
	13 weeks	*Faecalibacterium prausnitzii*	Decreased adipose tissue inflammation Lowered AST/ALT Increased fatty acid oxidation Improved intestinal integrity	Mice	[182]

*Abbreviations: CRP, C-reactive protein; DM, diabetes mellitus; Th1, T-helper 1 subtype; NAFLD, non-alcoholic fatty liver disease; PUFA/SFA, poly-unsaturated fatty acids / saturated fatty acids; IL, interleukin; TLR, toll-like receptor; NF-KB, Nuclear factor kappa-light-chain-enhancer of activated B cells.*

### 4.2. Probiotics

Probiotics are organic supplements that contain non-pathogenic live strains of bacteria, most commonly containing *Lactobacilli* and *Bifidobacteria* [183]. *Lactobacillus*, in particular, has significant roles in breakdown of glycans in the intestinal mucus layer, which not only serves to drive mucosal barrier regeneration, but also provides a nutrient source to other bacterial species which lack hydrolytic enzymes capable of digesting glycans into monosaccharides [184]. *Lactobacilli* outpace pathogenic bacteria by producing adhesins that allow binding to the host mucus layer [185]. Further, a study examining 58 different strains of lactic acid bacteria on a medium with cholesterol and bile acids, which simulated the human gastrointestinal tract environment, showed cholesterol reduction rates as high as 55% [164]. *Bifidobacterium* also plays significant roles in optimizing overall gut microbial composition. This is supported by recent findings showing that *Bifidobacterium animalis* supplementation significantly lessened the effects of dysbiosis induced by antibiotic administration. Further, *Bifidobacterium bifidum* probiotic supplementation in individuals with dyslipidemia for 3 weeks resulted in significantly decreased total cholesterol and LDL cholesterol [165]. As such, combination of these two main genera in probiotic supplements has been shown to positively impact several metrics of overall health, including the ability to lower serum levels of total cholesterol, LDL and TGs.

For example, administration of probiotics containing *Lactobacillus acidophilus*, *Lactobacillus casei* and *Bifidobacterium lactis* in hypercholesterolemic volunteers reduced LDL cholesterol by 10.4% and total cholesterol by 8.1% over a 10-week study period [166]. When examining the time that isolated LDL from serum samples took to oxidize, it was found that the probiotic group had a longer lag time in LDL oxidation, demonstrating the benefit of supplementation of these species in atherosclerosis development as well. Further, probiotics containing *Lactobacillus*, *Bifidobacterium* and *Streptococcus* were found to increase HDL cholesterol [163]. In addition, probiotics enhance bile acid profiles. For example, probiotic supplementation increased levels of *Lactobacillus* and *Akkermansia*, while reducing conjugated bile acids such as glycoursodeoxycholic acid and taurohyodeoxycholic acid [167]. Conjugated bile acids are reabsorbed through enterohepatic cycling and not excreted. Therefore, de novo bile acid synthesis is diminished, and less cholesterol is utilized. As such, probiotics can also facilitate excretion of bile acids through deconjugation reactions to increase utilization of cholesterol in de novo bile acid synthesis and improve states of dyslipidemia. 

### 4.3. Synbiotics

Synbiotics are formulations that combine prebiotics and probiotics. Prebiotics serve as substrates for probiotics that increase their beneficial effects on the gut microbiome as well as increase the likelihood of survival in the gut environment. These formulations are shown to improve metabolic parameters including lipid imbalance. In a 3-week study examining the effect of prebiotic xylo-oligosaccharides (XOS) alone or in combination with *Bifidobacterium animalis lactis* on gut microbiota composition, immune function, and serum lipid concentrations, Childs et al., showed increases in *Bifidobacteria* with XOS supplementation, significant increases in fasting HDL, and stimulation of Th1 and suppression of Th2 responses [168]. Further, *Lactobacillus acidophilus*, *Lactobacillus casei* and *Bifidobacterium bifidum* plus inulin significantly decreased concentrations of serum TG and VLDL over a 6-week period. In addition to *Bifidobacterium* and *Lactobacillus*, another probiotic, *Bacillus licheniformis* has been studied with XOS to determine symbiotic effects on gut dysbiosis in high-fat diet rats. The symbiotic combination of XOS and *Bacillus licheniformis* elicited reductions in serum LPS, a decrease in the F/B ratio, reductions in body weight and even reduced serum total cholesterol [169]. Therefore, combinations of prebiotics and probiotics can promote targeted changes in gut microbiota to improve dyslipidemia.

### 4.4. Gut Microbiota, Folate and Dyslipidemia

Folate is an essential molecule required for nucleotide biosynthesis and acts as a methyl donor for processes such as DNA methylation/epigenetic modification. It is well-established that probiotic bacteria in the gut microbiome are important for the production and synthesis of many compounds including vitamin K and the B vitamins. Though not all *Lactobacilli* and *Bifidobacteria* produce folate, specific species in these genera have folate-producing capabilities including *Lactobacillus. plantarum*, *Bifidobacterium adolescents*, and *Bifidobacterium pseudocatenulatum* [186]. However, the ability to produce folate is generally rare for gut microbiota, though most species are known to require folate for proper functioning [187]. As a result, folate therapy has been studied in the context of dyslipidemia through its targeted changes in gut microbiota. More specifically, folate has been shown to restore dysbiosis to beneficially alter serum lipids and cholesterol through its regulatory pathways. For example, Chen et al. studied different mice groups fed high-fat diets with or without folate supplementation and how it affected their body weight gain, fat distribution, and gut microbiota among other. In C57 BL/6J conventional (CV) mice fed a high-fat diet, supplementation with folate reduced weight gain, adipocyte sizes, dysbiosis, and down regulation of lipid-metabolism associated genes such as peroxisome proliferator-activated receptor-alpha (PPAR-alpha), hormone-sensitive lipase (HSL), and adiponectin [188]. In germ-free high-fat diet fed mice, folate supplementation had no effect on the overall body weight gain, whereas in CV high-fat diet fed mice there was a notable decrease in body weight gain, reduced expression of PPAR-alpha, HSL, adiponectin, and changes to short-chain fatty acid concentrations. This suggests that commensal bacteria in the CV microbiome employs folate to correct dysbiosis and improve host serum lipid profiles. 

Additionally, the *ADRB3* gene has surfaced as an important gene associated with obesity/dyslipidemia as it encodes for a beta-adrenergic receptor found on the surface of adipocytes and is important regulator of lipolysis. *ADRB3* gene methylation was negatively associated with serum folate levels in eutrophic adults but not overweight or obese adults [189]. This suggests that normal weight adults with adequate levels of folate experience lower levels of *ADRB3* gene methylation which can serve as a protective mechanism against obesity and altered lipid profiles. Further, folate-induced changes of gut microbiota are associated with sequelae of dyslipidemia including atherosclerosis and non-alcoholic fatty disease (NAFLD). For example, in a large study of 14,970 Chinese adults with hypertension, increased serum folate levels attenuated LDL and carotid intima-media thickness (CIMT) measurements, indicating an association between low folate levels and increased risk of atherosclerosis [171]. In addition, mice fed a high methionine low folate (HMLF) diet experienced higher fecal concentrations of *Porphyromonadaceae*, a family within *Bacteroidetes*, that has been associated with non-alcoholic fatty liver disease (NAFLD) [170], a condition characterized by insulin resistance and increased TG and LDL. Taken together, these findings support the beneficial effects of folate interventions and associated changes in gut microbiota in treating dyslipidemia, though more work is needed to examine the role of folate on the bacteria metabolism and the subsequent effects in dyslipidemia. 

### 4.5. Fecal Microbiota Transplantation in Restoring Dyslipidemia

Fecal microbiota transplant (FMT) is a treatment modality in which gut microbiota from a “healthy” individual is administered to a patient with the intent of restoring the gut’s microbial composition and providing a benefit to its overall function [190]. Currently used methods to administer fecal microbiota include via upper GI tract administration through oral capsules, nasogastric tube, and endoscopy or through the lower GI route via colonoscopy and rectal enema [190,191]. Although the only currently FDA-approved FMT therapy is for recurrent and refractory *Clostridoides difficile* infection, FMT has been proposed in recent years as a potential therapeutic intervention for patients with various pathologies including to improve lipid profile of patients with metabolic disease [192]. For example, FMT has been shown to induce rapid changes in gut microbiota composition profile and diversity, by increasing *Bifidobacterium* and decreasing sulfate-reducing bacteria including *Bilophila* and *Desulfovibrio* [173]. This resulted in improved serum lipid profiles, specifically total cholesterol, and triglycerides. A similar recent study supports the beneficial effects of FMT on improving lipid profile, which was associated with increased *Lactobacillus* and *Bifidobacterium* [174]. However, FMT in conjunction with lifestyle interventions including dietary modification more significantly reduced total and LDL cholesterol. As such, although beneficial effects of FMT have been demonstrated in several studies, combining treatments that promote targeted and generalized changes in gut microbiota may optimize outcomes. Few studies have also shown that FMT via oral capsules can also improve lipid metabolism by altering bile acids [175]. In addition to shifts in microbiota composition resembling the healthy donor microbiota, FMT also induced sustained decreases in fecal taurocholic acid and bile acid profiles that showed more similarity to donors. Therefore, bile acid-microbiota crosstalk can mediate some of the beneficial effects of FMT in dyslipidemia.

### 4.6. The Use of Faecalibacterium prausnitzii and Akkermansia muciniphila as Next-Generation Probiotics

In addition to *Lactobacillus* and *Bifidobacterium*, the two strains of beneficial gut bacteria strongly associated with improved lipoprotein profiles are *Akkermansia muciniphila* and *Faecalibacterium prausnitzii* [193]. They each constitute up to 5% of the total gut microbiome [194,195] in healthy individuals. Throughout the review, we have highlighted that the relative abundances of these species are ordinarily decreased in dyslipidemia, while pharmaceutical interventions and microbiota targeted therapies increase their concentrations. In as much as “precision probiotics” could be used to modulate specific microbiota-driven regulatory mechanisms involved in lipid metabolism, it is important to consider the inter-bacterial microbiota interactions and the resulting impact on the microbiota and the host. It is well known that bacteria act synergistically and collectively with other microbial species within our gut, therefore the effects of these next-generation probiotics on lipid metabolism may largely depend on the microbial interactome. However, establishing causality of inter-bacterial interaction network in impacting dyslipidemia, or in health and disease, in general, is a daunting task given the complexity of the factors involved. Nevertheless, manipulation of a single abundant gut commensal strain is one way in assessing changes in the gut microbiota metabolic pathways and how it impacts community structure and host responses [196]. In this regard, promising data has shown that *Akkermansia* and *Faecalibacterium* introduction, generally exerts favorable effects in states of dysbiosis seen in dyslipidemia. In the following subsections, we describe the potential mechanisms by which these two species exert their effects on improving lipid imbalance as well as their future applications as next-generation probiotics.

#### 4.6.1. *Akkermansia muciniphila*

*Akkermansia muciniphila* colonizes the gastrointestinal tract early in childhood and its population declines in the elderly [197]. *Akkermansia* has been shown to be positively and significantly correlated with PUFA/saturated fatty acid ratio, a marker associated with increased HDL cholesterol and negatively correlated with the onset of dyslipidemia and atherosclerosis [176]. Further, *Akkermansia muciniphila* interacts with the host immune system via an important outer membrane protein, named Amuc_1100, which has affinity for Toll-like receptor 2, thus preventing antigens such as LPS from binding and triggering an immune response [198]. By lowering metabolic endotoxemia induced from LPS-producing bacterial species and improving gut barrier integrity, current findings have shown that *Akkermansia* can reduce body fat mass and improve dyslipidemia [129] via restoration of HDL and LDL, an effect worsened by HFD-induced obesity.

It has also been shown that novel pharmaceutical agents and their anti-hyperlipidemic events may be largely related to the increase in *Akkermansia* [177]. For example, *Periplaneta americana* extract (PAE) increased gut concentrations of *Akkermansia muciniphila* to induce beneficial effects on oxidative stress and inflammation, commonly seen in metabolic disease [130]. This was shown through increase in tight junction proteins, zonulin-1 and occludin, that maintain gut barrier integrity and activation of interleukin-10 (IL-10) that mediate its effects through JAK1/STAT3 signaling pathways. Further, the therapeutic effect of another novel plant-alkaloid antihyperlipidemic agent, nuciferine, was directly correlated with *Akkermansia* enrichment [177]. Similar findings show that *Akkermansia* may exert these effects through its mucus degrading qualities, which provides other bacterial species with a consistent supply of nutrients which aids their survival [199]. This degradation of the mucous layer is carried out by glycosyl hydrolases releasing glycan oligosaccharides, promoting cross-feeding and increased richness and diversity in the microbial community. However, it is important to note that the relationship of *Akkermansia* on maintaining a healthy gut layer and preventing metabolic endotoxemia is dose-dependent, as over degradation of the mucin layer in combination with other mucin-degrading bacteria can increase gut permeability [200]. At the same time, a minimal concentration of *Akkermansia* is required for metabolic benefits to be present [178]. In patients with metabolic syndrome, *Akkermansia* comprised 0.08% of the total gut microbial abundance and it was not until *Akkermansia* reached 0.2%, that risk of metabolic syndrome decreased. Taken together, these findings provide strong evidence for *Akkermansia muciniphila* in restoring the unfavorable effects on intestinal barrier integrity, inflammation and decreased microbial richness seen in dyslipidemia.

#### 4.6.2. *Faecalibacterium prausnitzii*

Another potential next-generation probiotic is *Faecalibacterium prausnitzii* which has been identified as playing an important role in energy production and has an anti-inflammatory role which can combat chronic conditions such as dyslipidemia [179]. *Faecalibacterium prausnitzii* is shown to cross feed off other intestinal microbiota produced metabolites or dietary supplements to yield SCFAs, which produces most of its beneficial effects and it has been regarded as one of the most important butyrate producing bacteria within the human gut [201]. Butyrate has anti-inflammatory role via inhibition of nuclear factor kappa-light-chain-enhancer of activated B cells (NFκB) and therefore decreased interleukin 8 (IL-8), which causes chemotaxis neutrophils into tissues during inflammatory processes. Additionally, recent findings have shown that *Faecalibacterium*-derived butyrate can down-regulate TLR3 and TLR4 to reduce activity of the TLR-NFκB and HDAC axes [180]. Toll-like receptors are important receptors activated by LPS that induce inflammation via NFκB signaling, which is commonly elevated in metabolic diseases like dyslipidemia [202]. Further, *Faecalibacterium prausnitzii* stimulate mucin secretion thus maintaining intestinal barrier integrity [203]. These beneficial mechanistic effects of *Faecalibacterium prausnitzii* translate to improvements in lipid profile [181]. For example, oral introduction of *Faecalibacterium prausnitzii* increased adiponectin signaling and fatty acid oxidation, while reducing TG, phospholipids, and cholesterol esters [182]. These effects were associated with an overall improvement in gut microbial richness and diversity as well as improved intestinal integrity through stabilization of the mucous layer. Therefore, *Faecalibacterium prausnitzii* plays an important role in improving cholesterol imbalances commonly seen in various metabolic disease.

## 5. Conclusions and Perspective

While new studies continue to emerge, substantial evidence supports the involvement of gut microbiota and states of dysbiosis in the development and progression of metabolic diseases, such as dyslipidemia. Through advancements in our understanding of the role of gut microbiota and their metabolites in lipid metabolism, studies have shifted towards identifying ways to leverage its therapeutic potential into creating targeted microbiota therapies to improve lipid imbalances. Current first-line interventions include improved dietary intake through the MD and statin therapy that exert their positive effects through generalized, yet favorable changes, by increasing SCFA-producing bacteria like *Lactobacillus*, *Eubacterium*, *Faecalibacterium*, *Bifidobacterium* and *Akkermansia*, that are key in maintaining gut barrier integrity. Further, prebiotics, probiotics, synbiotics, FMT and next-generation probiotics provide a simple, yet effective treatment modality for dyslipidemia by enriching the gastrointestinal tract of affected individuals with beneficial microbial species. While research on gut microbiota and its therapeutic significance has yielded notable results, its clinical applications in humans have been limited. In order to draw causative conclusions between improvements in chronic diseases and use of microbiota modulators, thorough investigation is needed to understand the commensal, known as well as currently unknown, role prior to disease and throughout its progression. Thus, future directions of microbiota modulation treatments of dyslipidemia should include longitudinal studies that consider the inherent gut microbiota variations between individuals while investigating changes in microbiota profile prior, during, and after treatment. Although this is a time and labor-intensive process, the potential benefits of modulating gut microbiota for the treatment of dyslipidemia have been promising.

## Figures and Tables

**Figure 1 nutrients-15-00228-f001:**
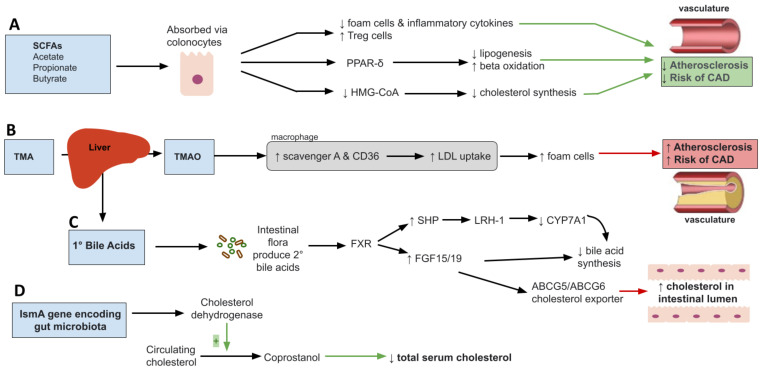
**Role of Gut Microbiota Metabolites on Dyslipidemia and Atherosclerosis.** (**A**) SCFAs are absorbed by colonocytes, leading to decrease foam cells, inflammatory cells, and decrease HMG-CoA thereby decreasing cholesterol synthesis. SCFAs increase T-regulatory cell activation and increase PPAR-δ activity, increasing beta-oxidation and decreasing lipogenesis. Increase SCFA production decreases atherosclerosis and risk of CAD. (**B**) Trimethylamine gets oxidized in the liver to form TMAO, which increases scavenger A and CD36 activity of macrophages and LDL uptake. This leads to formation of foam cells, contributing to increased atherosclerosis and CAD risk. (**C**) Primary bile acids synthesized by the liver are conjugated to secondary bile acids by gut microbiota. Secondary bile acids upregulate FXR leading to increase production of inhibitory nuclear receptor SHP. SHP interacts with living receptor homolog-1 to suppress transcription, interacts with CYP7A1 gene, decreasing synthesis of primary bile acid. FXR also induces FGF15 and FGF19 increasing ABCG5/ABCG6 cholesterol exporter, which helps translocate bile acids to the lumen. (**D**) IsmA encoding bacterial species contain enzymatic capability of converting circulating cholesterol to coprostanol which is excreted via feces, thereby decreasing serum cholesterol levels. Abbreviations: SCFA, short-chain fatty acids; Treg, T regulatory; CAD, coronary artery disease; PPAR, Peroxisome proliferator activated receptor; HMG-CoA, hydroxymethylglutarly Coenzyme A; TMA, Trimethylamine; TMAO, Trimethylamine *N*-Oxide; FXR, Farsenoid X receptor; SHP, small heterodimer partner; FGF, Fibroblast growth factor; LRH-1, living receptor homolog-1; ABCG; ATP-binding cassete gene; IsmA, intestinal sterol metabolism A; ↑, upregulated; ↓, downregulated.

**Figure 2 nutrients-15-00228-f002:**
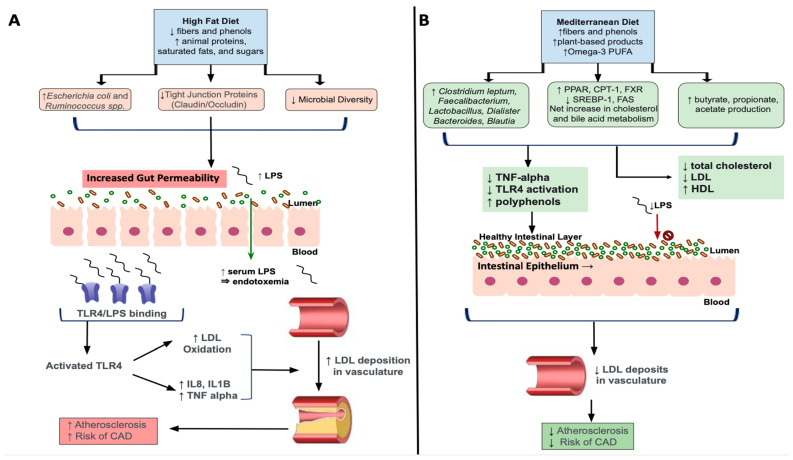
Effects of a High-Fat Diet and a Mediterranean Diet on Dyslipidemia and Atherosclerosis. (**A**) The High-Fat Diet is composed of large quantities of animal proteins, saturated fats and sugars with decreased fibers and phenols. HFD feeding increase Gram-negative bacteria *Ruminococcus* and *Escherichia* spp. while intestinal tight junction proteins occludin and claudin and microbial diversity are reduced. This leads to increased gut permeability, LPS enters the blood stream, resulting in metabolic endotoxemia. Lipopolysaccharides bind TLR-4 on circulating host cells to increase pro-inflammatory cytokines, increased inflammation and reactive oxygen species. Oxidized LDL builds up, causing plaques and increased risk of atherosclerosis. (**B**) The Mediterranean diet is rich in fibers and phenols, plant-based products, and omega-3 poly-unsaturated fatty acids. The MD increase *Clostridium leptum* and key genera *Faecalibacterium*, *Lactobacillus*, *Dialister*, *Bacteroides*, *Dialister*, *Bacteroides* and *Blautia.* It increases SCFA, while upregulating cholesterol and bile acid metabolism by increasing PPAR, CPT-1, FXR activity and decreasing SREBP-1 and FAS activity. This lead to decreased TNF-alpha, TLR4 activation, total cholesterol and LDL, increase HDL and supports intestinal barrier integrity. Less LDL builds up, lessening atherosclerosis and CAD risk. *Abbreviations:* LPS, Lipopolysaccharides; TLR4, Toll-like receptor 4; LDL, Low-density lipoprotein; IL, interleukin; TNF, Tumor necrosis factor; CAD, coronary artery disease; PUFA, poly-unsaturated fatty acids; PPAR, Peroxisome proliferator activated receptor; CPT-1, Carnitine palmitoyl transferase I; SREBP-1, sterol-regulatory element-binding protein 1; FAS, fatty-acid synthase; HDL, High-density lipoprotein.

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
