# Peer review of "Treatment of Dyslipidemia through Targeted Therapy of Gut Microbiota"

_nutrients, 2023, doi:10.3390/nu15010228_

Round 1

Reviewer 1 Report

Dear authors, congratulations on the review presented and the effort made to complete it. The main question addressed by the research [Treatment of Dyslipidemia through Targeted Therapy of Gut Microbiota] is relevant and exciting. It adds value to what has been published, especially regarding plausibility and elegance. The paper is well-written, straightforward, and easy to read. As the only objection, the conclusions seem too extensive,  and perhaps they should communicate the main idea in a more structured way by delivering truly crucial messages. 

Author Response

Thank you for your positive comments which helped improved our paper. As requested, we have revised the Discussion section to better obviate the main points of our review. In addition, at the request of Reviewer #2, we have incorporated additional evidence to several subsections, depicting the relationship between the gut microbiota, it metabolites and dysregulation of lipid metabolism.

Reviewer 2 Report

The authors explored the associations between dyslipidemia and intestinal flora dysbiosis, and then discussed the roles of key microbial metabolites (short chain fatty acids, primary bile acids and microbial cholesterol dehydrogenases) in the onset and progression of dyslipidemia. Afterwards, the authors described the improvement of abnormal lipid metabolism by adjusting a diet capable of targeting and regulating gut microbiota. They explored the several effectively ways on targeted microbiota therapy for management of dyslipidemia. Overall, the article is well organized, and its presentation is good, thus the paper is probably publishable.

However, there are still some issues need to be revised before acceptance:

(1) The association between gut microbiota and metabolic diseases, such as obesity, T2DM and NAFLD, has been in the spotlight for years. The authors summarized that the gut microbiota plays critical role in the onset and progression of dyslipidemia, but whether the changes of gut microbiota are the direct cause of abnormal lipid metabolism needs to be discussed in depth.

(2) In section 2.4 “Microbial Cholesterol Dehydrogenases”. In this part, the authors pointed out the importance of microbial enzymes in regulating cholesterol metabolism, and the authors should pay more attention to summarize more functions of microbial enzymes, which is beneficial for researchers to further elucidate the effects of gut microbiota on modulation dyslipidemia.

(3) In section 3 “Diet and its effects on gut microbiota and Dyslipidemia”, the authors claimed that the diet is the most important determinant in shaping the microbiota ecosystem, but they only discussed the effects of high-fat diet and Mediterranean diet, please explain the reason for choosing only two kinds of diet. Please consider the need for additional dietary structure on gut microbiota.

(4) Dietary and lifestyle changes may affect the structure, composition, and metabolic function of gut microbiota. The methods of altering the structure of gut microbiota mentioned by the authors in “Targeted Microbiota Therapies”, but the changes of colony structure are uncontrollable. Thus, it is debatable whether the methods mentioned in this article can target the regulation of gut microbiota to alleviate abnormalities in lipid metabolism.

(5) In section 4.6, the authors pointed out that the use of Faecalibacterium prausnitzii and Akkermansia muciniphila as next-generation probiotics, and summarized the functions of these two probiotics separately. The gut microbiota can’t play an independent role in regulating metabolic function, please consider whether there is synergy between microorganisms in regulation of lipid abnormalities.

(6) Please change the serial number of the title in line 705.

Author Response

Thank you for the constructive comments that helped improved our paper.

1) The association between gut microbiota and metabolic diseases, such as obesity, T2DM and NAFLD, has been in the spotlight for years. The authors summarized that the gut microbiota plays critical role in the onset and progression of dyslipidemia, but whether the changes of gut microbiota are the direct cause of abnormal lipid metabolism needs to be discussed in depth.

Response. Thank you for the comment. We have expanded this discussion by presenting more studies depicting the direct relationship between gut microbiota, its metabolites and lipid dysmetabolism (Line 85-104).

(2) In section 2.4 “Microbial Cholesterol Dehydrogenases”. In this part, the authors pointed out the importance of microbial enzymes in regulating cholesterol metabolism, and the authors should pay more attention to summarize more functions of microbial enzymes, which is beneficial for researchers to further elucidate the effects of gut microbiota on modulation dyslipidemia.

Response: We have included a paragraph presenting the role of some specific microbial enzymes involved in lipid metabolism and their associations with dyslipidemia (Line 452-465).

(3) In section 3 “Diet and its effects on gut microbiota and Dyslipidemia”, the authors claimed that the diet is the most important determinant in shaping the microbiota ecosystem, but they only discussed the effects of high-fat diet and Mediterranean diet, please explain the reason for choosing only two kinds of diet. Please consider the need for additional dietary structure on gut microbiota.

Response: We have focused on two main diets that have been consistently associated with groups of bacteria with shared functional role in metabolic pathways across healthy individuals and patients with metabolic disease states. As such, processed foods and animal-derived foods, that are characteristic of a Western diet, were consistently associated with higher abundances of Firmicutes, Ruminococcus species of the Blautia genus and endotoxin synthesis pathways. The opposite is true for plant foods and fish (characteristic of Mediterranean diet), which were positively associated with short-chain fatty acid-producing commensals and pathways of nutrient metabolism that confer protection and health benefits. We have briefly provided the rationale for focusing on these two diets (Line 481-483).

(4) Dietary and lifestyle changes may affect the structure, composition, and metabolic function of gut microbiota. The methods of altering the structure of gut microbiota mentioned by the authors in “Targeted Microbiota Therapies”, but the changes of colony structure are uncontrollable. Thus, it is debatable whether the methods mentioned in this article can target the regulation of gut microbiota to alleviate abnormalities in lipid metabolism.

Response: Thank you for your comment. We have included a statement, noting the role of other factors such as diet and lifestyle on the structure, composition and metabolic function of gut microbiota (Line 679-685).

(5) In section 4.6, the authors pointed out that the use of Faecalibacterium prausnitzii and Akkermansia muciniphila as next-generation probiotics, and summarized the functions of these two probiotics separately. The gut microbiota can’t play an independent role in regulating metabolic function, please consider whether there is synergy between microorganisms in regulation of lipid abnormalities.

Response: We have discussed the critical role of interbacteria interactions in regulation of lipid metabolism (Line 973-985).

(6) Please change the serial number of the title in line 705.

Response: This has been changed